# Pigment Production Improvement in *Rhodotorula mucilaginosa* AJB01 Using Design of Experiments

**DOI:** 10.3390/microorganisms9020387

**Published:** 2021-02-14

**Authors:** Alejandra Garcia-Cortes, Julián Andres Garcia-Vásquez, Yani Aranguren, Mauricio Ramirez-Castrillon

**Affiliations:** 1Facultad de Ciencias Básicas, Universidad Santiago de Cali, Campus Pampalinda, Calle 5 # 62-00, Santiago de Cali 760035, Colombia; alejandra111795@gmail.com (A.G.-C.); juliandres9502@gmail.com (J.A.G.-V.); 2Facultad de Ciencias Básicas y Biomédicas, Universidad Simón Bolívar, Carrera 59 # 59-65, Barranquilla 080003, Colombia; yani.aranguren@unisimonbolivar.edu.co

**Keywords:** Box-Behnken design, carotenoids, β-carotene, cellular stress, Plackett-Burman design, *Rhodotorula*

## Abstract

The discovery of biopigments has received considerable attention from the industrial sector, mainly for potential applications as novel molecules with biological activity, in cosmetics or if aquaculture food supplements. The main objective of this study was to increase the production of carotenoid pigments in a naturally pigmented yeast by subjecting the yeast to various cellular stresses using design of experiments. The fungal strain *Rhodotorula mucilaginosa* AJB01 was isolated from a food sample collected in Barranquilla, Colombia, and one of the pigments produced was β-carotene. This strain was subjected to various stress conditions, including osmotic stress using different salts, physical stress by ultraviolet (UV) light, and light stress using different photoperiods. The optimal growth conditions for carotenoid production were determined to be 1 min of UV light, 0.5 mg/L of magnesium sulfate, and an 18:6 h light/dark period, which resulted in a carotenoid yield of 118.3 µg of carotenoid per gram of yeast.

## 1. Introduction

Pigments are chemical substances that confer color to other substances through the optical effect of sunlight or are powders that, when mixed with liquids, can dye the surfaces of other materials. Pigments are a significant raw material in the textile, food, cosmetic, and clinical industries, where they are used as additives to improve certain products’ appearance and make them more attractive to consumers. Dyes and pigments are currently obtained from animal, plant, or mineral sources, as well as by synthetic (production of molecules using chemical reactions) and biotechnological (pigments produced by microorganisms) processes [1].

Carotenoids are organic pigments derived from isoprene, formed by a 40-carbon chain characterized by the presence of a long polyene chain, in which the number of double bonds may vary from 3 to 15, which is responsible for the production of the colors perceived by the human eye. Carotenoids feature yellow, orange, and red colors, which are widely distributed in microorganisms and plants. Carotenoids are highly valuable in the pharmaceutical, cosmetic, food, and animal feed industries not only because they serve as vitamin A precursors but also because of their coloration, antioxidant properties, potential tumor-inhibiting activity, and the immune response they induce, which can protect against bacterial and fungal infections [2].

Plant extracts, particularly β-carotene and xanthophylls, have been used for over five decades as sources of carotenoids [3]. However, various microbiological processes have been commercially exploited for the production of these compounds [4]. Industrial carotenoid pigments such as β-carotene and astaxanthin are used as natural food coloring agents or as aquaculture feed additives [5]. Owing to the fact that most of the current chemically produced carotenoids can cause side effects in atopic patients, including clinical symptoms such as dermatitis, rhinitis, asthma, hives, and angioneurotic edema, researchers are exploring other ways to obtain these molecules from natural products, such as plants and those obtained from microbial sources [6].

Carotenoids are mainly produced by filamentous fungi, yeast, and some bacterial, algal, and lichen species. Among the microbial sources of carotenoids, in addition to the algae *Dunaliella* sp., two yeast species of commercial interest, *Xanthophyllomyces dendrorhous* and *Rhodotorula glutinis,* have been identified. Among the microorganisms exploited for carotenoid production are yeasts used for specific industrial applications, such as *X. dendrorhous* for astaxanthin production, which has a globally established market [7]. Moreover, some *Rhodotorula* species, such as *R. rubra* and *R. glutinis,* produce β-carotene, torulene, and torularhodin. Torularhodin has the highest oxidation level among carotenoids in low-nitrogen conditions for yeast development [8]. Although *R. glutinis* produces carotenoids in various proportions, the β-carotene content is relatively low among wild strains [9]. In fact, has been reported that main carotenoid for *R. mucilaginosa* is torularhodin [10]. 

Biopigments are increasingly used in applied research as a result of the need to eliminate the use of chemically produced pigments not only because of human health concerns but also because of their environmental impact. In Colombia, research conducted by wastewater treatment plants reported that effluents contain high levels of chemical substances (e.g., colorants) that are hard to eliminate and highlight the need to develop new processes or protocols to completely eliminate these pollutants from water [11].

One of the main advantages of using synthetic pigments is their high yield at low cost. However, their high toxicity to humans and harmful environmental effects, such as contamination of water bodies, have spiked interest among researchers to develop new pigment types. The demand for pigments produced by microbes has been increasing because these pigments can be obtained using low-cost and low-maintenance growth media. Several reports showed independent or multivariate effect of culturing conditions to increase pigment production, such as temperature, pH or carbon source [10,12]. However, the increasing of pigment production through cell stress by physical and chemical agents were not assessed in a multivariate approach. Accordingly, the main objective of this study was to increase the yield of carotenoids by subjecting a yeast isolated from Barranquilla, Colombia, to osmotic, physical, and light stress, using design of experiments.

## 2. Materials and Methods

### 2.1. Yeast Isolation

To isolate pigmented yeasts, samples were collected from wastewater, crab and fish intestines, soil, and food sold in street carts in Barranquilla, Colombia. Barranquilla is located in the Atlantico Department in northern Colombia in South America and is located on the western shore of the Magdalena River and is 15 km away from the river mouth in the Caribbean Sea. Barranquilla has a tropical dry climate, characterized by dry vegetation and a mean temperature of 27 °C. Barranquilla features a dry and a wet season, which facilitates the isolation of various microorganisms capable of enduring a wide range of environmental conditions, ranging from dry to extremely wet conditions.

Samples were macerated and used to create serial dilutions (10^−1^ to 10^−6^) in YPG broth (glucose 20 g/L, yeast extract 5 g/L, and peptone 10 g/L) and were inoculated on YPG agar plates (bacteriological agar 20 g/L). The inoculated agar plates were incubated for 5 days at a temperature of 27 °C. Following isolation, the pigmented yeasts were maintained in Luria-Bertani medium (tryptone 10 g/L, yeast extract 5 g/L, NaCl 10 g/L, and agar 20 g/L), neutral pH YPG agar, and YPG agar with pH adjusted to 5.5. The cultures were incubated for 48 h at 27 °C. Finally, each strain was cryopreserved in YPG broth and glycerol 50% *v/v* solution at −20 °C.

### 2.2. Molecular Identification

Genomic DNA was extracted using the GeneJet kit (Thermo Fisher, Waltham, MA, USA) using a modified protocol. A standard inoculum of the AJB01 strain was prepared and adjusted to a concentration of 1 × 10^8^ cells/mL in YPG broth. All subsequent steps followed the GeneJet kit protocol. SYBR Green 1% (*w/v*) agar gel electrophoresis was used to assess the quality of the extracted genomic DNA.

For amplification of the ITS (ITS1-5.8S-ITS4) and LSU (large ribosomal subunit 26S) regions, a 25-µL PCR mastermix was prepared using 20 pmol/µL of primers ITS1 and ITS4 (Macrogen, Seoul, Korea) or NL1 and NL4 (Macrogen), 25 mM MgCl, 10 µM dNTPs, 10× buffer, 2.4% DMSO (*v/v*) and 0.2 U Taq polymerase. After the PCR reaction, amplified products were run on a 1.5% (*w/v*) agarose gel to visualize the presence of 500–600-bp products. Finally, Sanger sequencing of the PCR products was outsourced at CORPOGEN (Colombia). For RPB2 (second largest subunit of the RNA-polymerase II gene) sequence comparison, we picked the gene from whole annotated genome. The sequences were edited and compared to existing sequences deposited at GenBank and Mycobank. A threshold match ≥98.41% (ITS) and ≥99.51% (LSU) was used for species level identification [13]. A phylogenetic tree was constructed for RPB2 gene in clade *Rhodotorula* to classify AJB01 strain. The method was Maximum Likelihood (ML) with bootstrap of 1000 replications. The best model of nucleotide substitution and phylogenetic tree were estimated with software MEGA X.

### 2.3. Biomass Determination

To determine the Dried Weight Cells (DWC) we distributed the biomass from each Petri dish in two tubes. Half of the biomass was used for pigment extraction (Section 2.4) and the other half for biomass determination. The microcentrifuge tube for biomass determination was centrifuged at 8000 rpm for 5 min twice and resuspended in sterile distilled water. The final pellet was dried in a drying oven for 48 h and weighted in an analytical balance. The difference between weights of the microcentrifuge tubes with the pellet and without pellet was considered the biomass in grams. 

### 2.4. Pigment Extraction

Biomass collected from each plate grown for 120 h was standardized to 0.8 g and suspended in 3.5 mL of distilled water in 12-mL conical tubes, which were then centrifuged at 3000 rpm for 5 min, and the supernatant was discarded. The pellet was then suspended in 1 mL of dimethyl sulfoxide (DMSO), and the tubes were placed in a water bath at 55 °C for 1 h, with vortex every 10 min. The tubes were centrifuged at 3000 rpm for 5 min, and the supernatant was transferred to a chilled Falcon tube, which was covered with aluminum foil and stored at 0 °C. This step was repeated twice, after which the pellet was suspended in 1 mL of acetone at 5 °C and vortexed for 15 min. The tube was then centrifuged at 3000 rpm for 5 min, and the supernatant was transferred to the chilled tube containing pigments with DMSO. This step was repeated until no pigments were visible in the pellet or once the pellet started to detach from the tube. Following pigment extraction from the pellet, 1.5 mL of saturated NaCl solution and 1 mL of hexane were added to the chilled tube, which was then vortexed for 1 min. The chilled tube was then centrifuged for 5 min, and the supernatant (hexane-containing pigments) was transferred to a clean tube covered with aluminum foil. The tubes were stored at 0 °C in the dark.

To determine the relationship between dry biomass and pigment production, the pigments were quantified using spectrophotometry with a hexane blank. Finally, the total carotenoid content (µg/g) for each extraction was estimated using Equation (1) [14]:(1)$$Totalc\;arotenoids=\frac{OD*Volume\left(\mathrm{mL}\right)\;*\;10^{4}\frac{\mu g}{\mathrm{mL}}}{EC\;*\;Biomass\left(g\right)}$$
where *OD* = Optical density at 485 nm; Volume = volume in mL; *EC* = Extinction coefficient of hexane = 2592; *Biomass* = Dried weight of the sample in g.

### 2.5. Univariate Analysis of NaCl Stress

The AJB01 strain was cultured in YPG broth at increasing NaCl concentrations: 0.2 M, 0.4 M, 0.6 M, 0.8 M, 1 M, 1.2 M, 1.4 M, and 1.6 M). Yeast growth was determined by the turbidity method (absorbance at 600 nm), and pigment production was quantified at 485 nm. A comparative growth curve was plotted to determine the optimal NaCl concentration compared with a control grown without NaCl. Two strains, AJB01 and AJB02, were selected and monitored in YPG growth media. To evaluate the growth kinetics of these strains, cells were periodically counted using a Neubauer chamber.

The initial concentration of the inoculum was 1 × 10^8^ cells/mL, and 10 mL was used as inoculum for each strain. To determine the yeast generation time and cell viability in media, cells were counted every hour for AJB01 strain. Simultaneously, 2-mL aliquots were collected every 12 h for 120 h and were stored throughout the week in previously sterilized tubes and dried and weighted using an analytical scale (ME204 model, Mettler Toledo, Collumbus, OH, USA). Aliquots were centrifuged, and the pellet was dried in an oven at 70 °C until constant weight was recorded. Finally, the pellet’s weight was recorded using an analytical scale. The calculated weight difference was reported in g/L. All experiments were conducted in technical triplicate.

### 2.6. Screening for Other Stress Factors Using Plackett–Burman Design

A Placket–Burman experimental design was used to evaluate the effect of different types of cellular stress on AJB01 strain biomass and carotenoid production. The experimental design included 11 combinations to induce cellular stress. The components of MMS broth are glucose 1% *w/v*, yeast extract 0.1% *w/v*, (NH_4_)_2_SO_4_ 0.2% *w/v*, KH_2_PO_4_ 0.2% *w/v*, MgSO_4_·7H_2_O 0.05% *w/v*, and Cl_2_Ca·2H_2_O 0.0075% *w/v* [14]. LED lamps (6 W; 500 lumens), placed 25 cm above the cultures were used to control photoperiods and germicidal lamps (30 W; wavelength, 256 nm) were used to subject the cultures to ultraviolet (UV) radiation. Temperatures were controlled using refrigeration chambers. Three technical replicates were performed for each treatment, which was run for 96 h. At the end of the experiment, 0.80 g of yeast biomass was collected for pigment extraction, from which biomass (g/L) and carotenoid yield (µg/g) were calculated. Stress factors were statistically evaluated to determine the optimal growth conditions and therefore maximize carotenoid production.

### 2.7. Box–Behnken Experimental Design

The STATISTICA Academic software, version 13 (TIBCO Software Inc., Palo Alto, CA, USA), was used to generate a Box–Behnken 3^3^ experimental design to assess the following factors: magnesium sulfate concentration (0 to 1% *w/v*), UV light exposure time (0 to 2 min), and photoperiods (12 h:12 h to 24 h:0 h of light/dark) combined into 15 different treatments. A pre-inoculum standard was prepared using the AJB01 strain at a concentration of 4.6 × 10^7^ cells/mL, which was used to inoculate each of the 15 treatments. Successful inoculation was confirmed by counting the cells in a Neubauer chamber after 48 h of growth at 27 °C. The responses of the dependent variables, biomass (g), and carotenoid yield (µg/g), were evaluated.

## 3. Results and Discussion

We obtained 10 pigmented yeasts from all samples analyzed (Appendix A): two were grown from crab samples and eight from food samples. From these ten pigmented yeast strains, two were selected for subsequent experiments: AJB01 and AJB02. The other eight yeast strains were discarded as they had similar macro- and microscopic characteristics. From initial trials using the AJB01 and AJB02 strains, the first was found to be the most viable, growing in saline media, which was considered the first qualifying factor (i.e., osmotic cellular stress) to select strains for the subsequent experiments. The ITS and LSU regions (Genbank accessions MT889967, MT889968) were compared using BLAST and matches above 99% sequence similarity with the type strain sequences. However, species level identification could not be achieved due to *R. mucilaginosa* and *R. alborubescens* have almost identical sequences on these regions. We constructed a phylogenetic tree using the RPB2 gene, classifying and confirming AJB01 as *Rhodotorula mucilaginosa* (Appendix A). Pigmented yeast can be isolated from acidic, glacial, and seawater samples [15]. Moreover, *Rhodotorula* sp. has been isolated from dairy and meat products as well as from water and soil samples [16].

One of the stress factors considered was the saline concentration that AJB01 strain was capable of withstanding and using to overproduce biomass and carotenoids. Therefore, its performance in this yeast strain was studied in broth with various NaCl concentrations. NaCl was selected because of its relatively low cost and availability to be used at a large scale. AJB01 strain showed optimal growth at a salinity of 0.2 M (Appendix A), whereas AJB02 strain was entirely inhibited across the NaCl gradient, exhibiting no growth, and was therefore excluded from the cellular stress experiments.

In general, yeast biomass and metabolite production are linked to stress conditions in terms of dissolved oxygen, osmotic pressure, temperature, pH, and presence of metabolites such as ethanol [17]. Owing to high salt concentrations, osmotic stress can negatively or positively affect cellular functions. Loss of turgor pressure and lysis caused by metabolic sodium ion toxicity are some of the negative effects, whereas the positive effects include cellular adaptations to conditions requiring induction to osmoregulatory mechanisms, which, in the case of pigmented yeasts, generate metabolites to prevent cell lysis or damage [18,19]. Yeasts, being unicellular organisms, rely on physiological and composition change mechanisms to withstand stress conditions such as osmotic pressure and display a wide range of metabolic responses [20]. In certain cases, the yeast’s growth rate can decrease, whereas in other cases, the metabolic responses can result in an increased growth rate or production of osmoprotectant metabolites such as pigments. In fact, a previous study found that NaCl significantly increases the concentration of carotenoids such as β-carotene in *Sporidiobolus pararoseus* [21], *R*. *glutinis*, *R. mucilaginosa*, and *R. gracilis* and decreases the concentration of other pigments, such as torulene and torularhodin [22]. Comparative growth kinetics of the AJB01 strain cultured with and without 0.2 M NaCl (control, Appendix A), suggested that at a 0.2 M NaCl concentration (11.69 g/L), biomass production was higher than in the YPG broth control. Other researchers have reported that lower NaCl concentrations (0.5–2 g/L) did not significantly affect biomass production of *X. dendrorhous* (formerly *Phaffia rhodozyma*) [23], which indicates that only higher salt concentrations, such as those assessed in the present study, result in osmotic stress. When exposed to various types of stress (e.g., osmotic stress), yeasts are capable of activating metabolic pathways that result in increased growth rates and consequently larger biomasses.

After selecting the AJB01 strain, the factors that most influenced biomass and carotenoid production were evaluated. The Plackett–Burman design shows the different combinations of each treatment in addition to biomass concentration, estimated carotenoid concentration, and pigment production as response variables (Table 1).

The two factors that mostly affected carotenoid production were UV radiation and presence of MgSO_4_, with significant effects with regard to the statistical model value (*p* < 0.05) (Table 2). Exposure to UV radiation had a positive effect on carotenoid production in AJB01 strain, whereas MgSO_4_ had a negative effect. The increased pigment production by the strain when exposed to UV radiation could represent a protective mechanism, whereby the pigments protect the yeast cell from UV radiation, free radicals, enzymatic lysis, or high temperatures [24]. In contrast to our results, MgSO_4_ has been reported to increase pigment production up to 1.7× in *R. glutinis* [12].

We obtained optimal values for each cellular stress factor to maximize carotenoid yield of the AJB01 strain, which enabled us to establish a growth media to maximize pigment production. This does not include the exogenous factors temperature, UV radiation, and photoperiods. The proposed growth media provides a source of carbon and nitrogen and contains 1% *w/v* glucose, 0.1% *w/v* yeast extract, 0.1% *w/v* (NH_4_)_2_SO_4_, 0.2% *w/v* NaCl, 0.2 M, KH_2_PO_4_, and 0.0075% *w/v* CaCl_2·_H_2_O. These ingredients are essential to maximize carotenoid production in AJB01 strain, likely because the yeast may produce photoinducible compounds as a photoprotective mechanism against oxidative stress [25]. In contrast, certain salts, such as ammonium sulfate and magnesium sulfate, are required in low concentrations because they induce osmotic stress driven by cation exchange. Therefore, active transport is prioritized through cytoplasmic activation, in addition to other mechanisms, to preserve cellular structure, which could be destabilized by these salts [26] and result in decreased pigment production.

Considering the previous results, a Box–Behnken predictive model was established, including UV radiation (min), photoperiods (h), and magnesium sulfate percentage (*w/v*) as factors and total carotenoids and biomass as response variables (Table 3).

UV radiation and photoperiods are the exogenous factors that mostly affect carotenoid production (*p* < 0.05) (Table 4). On the other hand, magnesium sulfate did not significantly affect pigment production (*p* = 0.3). Moreover, only UV radiation significantly affected biomass production. Ultraviolet radiation triggers a defense mechanism against oxidative stress through the cell’s production of photo-protectant molecules, such as carotenoids and mycosporines [27]. The regression coefficient obtained was R^2^ = 0.97 for the model (Equation (2)), suggesting a strong fitness between the observed and predicted values.
Carotenoids = − 160.5008 – 103.8812 × UV (min) + 102.6566 × UV^2^ + 27.7829 × Photoperiod (hours of light) – 0.8453 × Photoperiod^2^(2)

Using Equation (2), response surface analysis plots were created, which suggest that carotenoid production correlates positively with exposure to UV radiation, as approximately 350 µg/g of carotenoids are produced after 2 min of UV exposure (Figure 1), compared to previous studies of *Rhodotorula mucilaginosa*, which produced between 82 and 184 µg/mL of carotenoids under stressful conditions [28]. However, magnesium sulfate did not significantly affect pigment production, and MgSO_4_ concentration around 1 g/mL resulted in ~40 µg/g of carotenoids.

Figure 1b shows how photoperiods and UV radiation exposure time affect carotenoid production. Carotenoid yield ranged from <10 µg/g to > 300 µg/g and was highest after 2 min of UV radiation, with a carotenoid yield of 300 µg/g. β-Carotene has been linked to response mechanisms and acts as a photoprotectant against reactive oxygen species, which accelerate cellular aging and may cause cellular lysis [29], increasing the transcription of genes associated with carotenoid production, such as in the case of *R. toruloides* [29]. This may explain the increased carotenoid production observed, as a photoprotectant, of the AJB01 exposed to UV radiation. Although photoperiod has been found to promote carotenoid production under extensive and or intense lighting exposure [30], in this study, the more aggressive wavelength of the UV light (256 nm) and its effect on nitrogenous bases may account for the stronger response of the AJB01 strain to UV exposure.

The effect of photoperiod and magnesium sulfate on biomass was relatively low (<0.07 g/mL and 0.09 g/mL, respectively). Since light affects the key genes involved with carotenoid production and their expression in *R. toruloides*, these results may suggest that the *R. mucilaginosa* AJB01 strain may be responding in a similar manner [31]. On the other hand, magnesium sulfate did not significantly affect biomass production (<0.07 g/mL). Magnesium sulfate is not a stressor for yeasts, and it stimulates fatty acid synthesis for membrane regulation and takes part in cellular stability [32]. Similarly, the lowest biomass (<0.1 g/mL) was observed when the AJB01 strain was expose to high MgSO_4_ concentrations and long UV radiation exposures. Although the stressors assessed in this study did not significantly affect biomass production [12], our model suggests that low MgSO_4_ concentrations and short UV radiation exposure times result in the highest yield > 0.08 g/mL.

Based on our model, the optimal UV radiation time, photoperiod length and MgSO_4_ taking into account both biomass and carotenoid production at the same time (Figure 2). To maximize carotenoid yield, we suggest exposing the AJB01 strain to UV radiation for 1.5 min since, at longer exposure times, the yeast is photo-activated and increases the production of photoprotectants, including carotenoids and other pigments, to reduce the adverse effects of UV radiation. It is advised to prepare growth media with low MgSO_4_ concentrations, because it did not significantly affect carotenoid production. The optimal photoperiod was found to be 18 h light. Although the wavelengths emitted by white LEDs cover a wide visible range (400–760 nm), these wavelengths are not as aggressive as those of UV radiation and therefore do not affect pigment production as much. Moreover, the desirability plot suggests that these factors do not promote biomass production, indicating that MgSO_4_ concentration, UV radiation exposure time, and photoperiod should be minimal.

To validate the model, β-carotene was quantified from the AJB01 strain exposed to the treatments recommended by the desirability plot (Figure 2) and compared it to a control treatment (strain without stressors). β-carotene was higher in the treatment than in the control (2.8 vs. 0.6 µg/mL). Total carotenoids were quantified as 118.3 µg/g in the treatment sample compared to 41.9 µg/g in the control. According to our model, the expected yield was 140.87 ± 4.23 µg/g under optimal conditions compared to 51,17 ± 1,54 µg/g for the control. Thus, other yields reported for *R. mucilaginosa* (110.8 µg/g and 117 µg/g) [32] cultured in different growth media were lower than the ones reported in this study as a result of cellular stress.

The results from our model suggest that these models do not consider errors associated with pigment extraction methods, as this model calculates total carotenoid production assuming that all cells were lysed, and carotenoids were fully extracted. Alternatively, slight variations may have taken place when implementing the extraction technique, which may be complex to solve in a laboratory setting. This highlights the importance of optimizing extraction methods to maximize the extraction of all pigments produced by the AJB01 strain.

Future studies should explore methods and analysis for other pigments, such as mycosporine, which is produced by *Rhodotorula mucilaginosa* subjected to UV radiation [33]. Mycosporine acts as a photoprotectant capturing wavelengths emitted by UV rays (265 nm), which are harmful to yeasts, resulting in the deamination of nitrogenous bases and preventing the replication of genetic material [27].

## 4. Conclusions

Two pigmented yeast strains were obtained from samples, one of which was identified as *R. mucilaginosa* and was selected for further experiments. The optimal treatment to maximize the production of pigments by the *R. mucilaginosa* AJB01 strain was UV radiation exposure for 1.5 min and a photoperiod of 18:6 light: dark hours. Although the highest carotenoid yield recorded was 350 µg/g, the optimal conditions also considering biomass production resulted in a carotenoid yield of 118.3 µg/g. The AJB01 strain can adapt to different growth conditions, including conditions that are lethal to other microorganisms, such as UV radiation exposure for more than 2 min, when this strain produced more carotenoids.

## Figures and Tables

**Figure 1 microorganisms-09-00387-f001:**
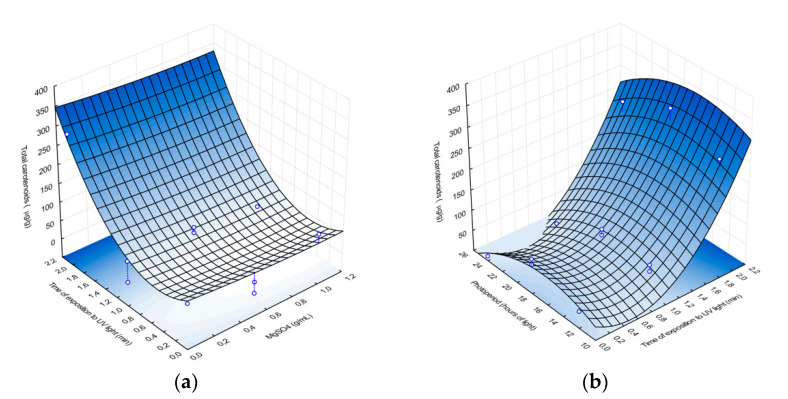
Effect of MgSo_4_ vs. UV exposure (**a**) and photoperiods vs. UV exposure (**b**) on carotenoid production.

**Figure 2 microorganisms-09-00387-f002:**
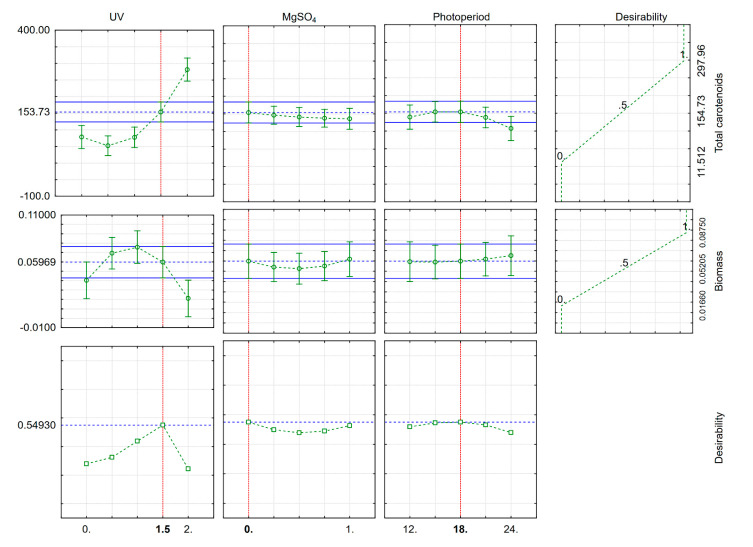
Desirability plot highlighting the optimal cellular stress parameters to maximize biomass and carotenoid production.

**Table 1 microorganisms-09-00387-t001:** Screening generated by the experimental design Plackett Burman.

Treatment	UV (min)	Temp (°C)	Fotoperiods (Hours of Light)	MgSO_4_ (% *w*/*v*)	NaCl (% *w*/*v*)	(NH_4_)_2_SO_4_ (% *w*/*v*)	Yeast Extract (% *w*/*v*)	Biomass (g)	Carotenoids (µg/g)	Culture Medium
**1**	0.0	4.0	12.0	1.0	0.2	0.2	0.1	0.105	30.830	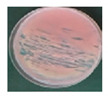
**2**	2.0	4.0	12.0	0.0	0.0	0.2	0.5	0.061	72.440	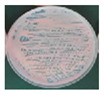
**3**	0.0	28.0	12.0	0.0	0.2	0.0	0.5	0.126	35.580	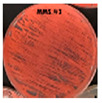
**4**	2.0	28.0	12.0	1.0	0.0	0.0	0.1	0.059	54.140	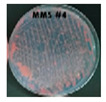
**5**	0.0	4.0	24.0	1.0	0.0	0.0	0.5	0.138	32.860	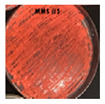
**6**	2.0	4.0	24.0	0.0	0.2	0.0	0.1	0.023	151.300	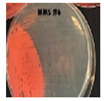
**7**	0.0	28.0	24.0	0.0	0.0	0.2	0.1	0.119	56.016	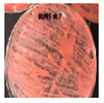
**8**	2.0	28.0	24.0	1.0	0.2	0.2	0.5	0.204	38.240	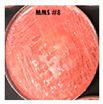
**9**	1.0	16.0	18.0	0.5	0.1	0.1	0.3	0.114	43.509	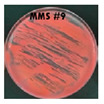
**10**	1.0	16.0	18.0	0.5	0.1	0.1	0.3	0.109	48.066	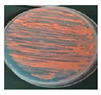
**11**	1.0	16.0	18.0	0.5	0.1	0.1	0.3	0.164	32.880	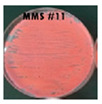

**Table 2 microorganisms-09-00387-t002:** ANOVA results of the different cellular stress factors analyzed using a Plackett–Burman design.

Factor	SS	df	MS	*F*	*P*
UV	3233.08	1	2333.082	12.35081	0.039075
Temperature	1337.77	1	1337.767	5.11045	0.108863
Photoperiod	911.88	1	911.879	3.4835	0.158817
MgSO_4_	3170.82	1	3170.819	12.11296	0.040045
NaCl	205.06	1	205.064	0.78337	0.441297
(NH_4_)_2_SO_4_	728.91	1	728.907	2.78452	0.193771
Yeast extract	1600.75	1	1600.745	6.11506	0.089836
Error	785.31	3	261.771		
Total SS	11973.58	10			

**Table 3 microorganisms-09-00387-t003:** Total carotenoids predicted using a Box–Behnken 3^3^ design.

Treatment	UV (min)	MgSO4 (% *w/v*)	Photoperiod(Hours of Light)	Carotenoids (Observed) (µg/g)	Predicted Values	Residual Values	Culture Medium
1	0	0	18	72.006	78.576	–6.569	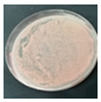
2	2	0	18	297.958	281.440	16.518	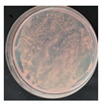
3	0	1	18	85.051	60.834	24.217	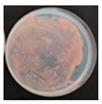
4	2	1	18	229.533	263.698	–34.165	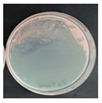
5	0	0.5	12	41.3692	51.163	–9.794	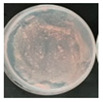
6	2	0.5	12	247.725	254.027	–6.302	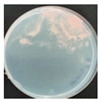
7	0	0.5	24	11.511	19.364	–7.853	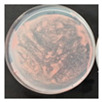
8	2	0.5	24	246.178	222.229	23.949	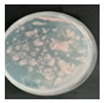
9	1	0	12	69.758	62.818	6.940	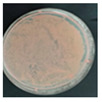
10	1	1	12	54.232	45.076	9.155	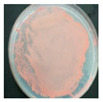
11	1	0	24	14.129	31.019	–16.889	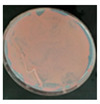
12	1	1	24	14.071	13.278	0.793	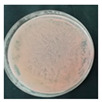
13	1	0.5	18	63.711	64.472	–0.760	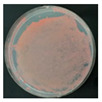
14	1	0.5	18	78.106	64.472	13.634	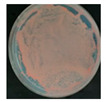
15	1	0.5	18	51.599	64.472	–12.873	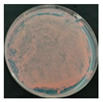

**Table 4 microorganisms-09-00387-t004:** Analysis of variance Box–Behnken design.

Factor	SS	df	MS	*F*	*P*
UV L + Q	121218.6	2	60609.32	134.2857	0.000001
MgSO_4_ L + Q	688.8	2	344.41	0.7631	0.297384
Photoperiod L + Q	5442	2	2720.98	6.0286	0.025309
Error	3610.8	8	451.35		
Total SS	132943	10			

## Data Availability

Not applicable.

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
