# Peer review of "Pigment Production Improvement in Rhodotorula mucilaginosa AJB01 Using Design of Experiments"

_microorganisms, 2021, doi:10.3390/microorganisms9020387_

Round 1

Reviewer 1 Report

The manuscript presented for review "Pigment production improvement using Design of Experiments in Rhodotorula mucilaginosa AJB01, a red yeast isolated from Barranquilla, Colombia" presents research on isolation and identification of Rhodotorula yeast, as well as the production of pigments by these yeasts.

The application of statistical experimental planning to these studies is interesting, it is a less common method, and allows to reduce the cost of the experiment, and the determined model can be used to optimize the research.

Specific comments and questions that arise when reading the manuscript (some of them are answered in later chapters, but it would also be useful to start with, which would make the text easier to understand):

Can Rhodotorula yeast pigments be used in industry, especially in food? These yeasts have no GRAS status

line 96 - why was it inoculated on LB?

Point 2.2 - How many strains have been identified? Only AJB01?

Point 2.4 - what is this AJB02 strain?

line 184 - additional RPB2 regions appeared, which was not described in the methodology, please correct it.

Table 1 - reduce the font of the header so that the names are in one line, MgSO4 lacks subscript

line 240 - Cl2Ca.H2O?

Table 3 - it would look better if the photos were of the same size as in table 1

Author Response

The manuscript presented for review "Pigment production improvement using Design of Experiments in Rhodotorula mucilaginosa AJB01, a red yeast isolated from Barranquilla, Colombia" presents research on isolation and identification of Rhodotorula yeast, as well as the production of pigments by these yeasts.

The application of statistical experimental planning to these studies is interesting, it is a less common method, and allows to reduce the cost of the experiment, and the determined model can be used to optimize the research.

Specific comments and questions that arise when reading the manuscript (some of them are answered in later chapters, but it would also be useful to start with, which would make the text easier to understand):

Can Rhodotorula yeast pigments be used in industry, especially in food? These yeasts have no GRAS status

A// Yeast carotenoids has big potential to be used in industry, including food industry. This is supported for several authors (Mata-Gomez et al. 2014, Frengova & Beshkova 2009, Kot et al. 2018). In fact, Astaxanthin is one example of industrial use in animal feeding (mainly aquaculture). Besides Rhodotorula have no GRAS status, the methodology implies to extract carotenoids from cultures, killing the cells. We need deep research in order to answer if Rhodotorula cells could be used directly in food industry, without pigment extraction.

line 96 - why was it inoculated on LB?

A// We used a standardized culture medium used in Microbiology Laboratory at Universidad Simón Bolivar (Barranquilla, Colombia), to grow and maintain microorganisms. As the composition of the Luria-Bertani medium is very similar to others nutritive culture medium, specific to yeasts, we did not have problems to grow the strain AJB01 in these conditions.   

Point 2.2 - How many strains have been identified? Only AJB01?

A// Yes, only AJB01 was identified.

Point 2.4 - what is this AJB02 strain?

A// AJB02 was another strain with different morphotype compared to AJB01. However, AJB02 did not grow with NaCl (line 194). For this reason, we discarded this strain.

line 184 - additional RPB2 regions appeared, which was not described in the methodology, please correct it.

A// We added RPB2 comparison in our experiments to full determine the species inside complex R. mucilaginosa/alborubescens. Both regions ITS and LSU were not enough to separate the species. In our case, we are working in a parallel experiment with the whole genome of the same strain. . So, we annotated the genome using the pipeline of mycocosm (JGI genome database) and extracted the RPB2 gene from the genome. In brief, we extracted the whole genome using the Omega EZNA Fungal DNA kit, and sequenced by Novogene using Illumina platform (paired-end) with 80X of coverage. After, we trimmed the reads using FastQC and Dynamictrim, and assembled the genome using Spades. The quality of the assembled genome was assessed with Quast. The genome annotation was done using the pipeline of Mycocosm. Because the extension of this methodology, we agree in not to write all methodology to extract this gene using genome sequencing.

Table 1 - reduce the font of the header so that the names are in one line, MgSO4 lacks subscript

A// Corrected.

line 240 - Cl2Ca.H2O?

A// Corrected to CaCl2.H20

Table 3 - it would look better if the photos were of the same size as in table 1

A// We increased the size of pictures.

Reviewer 2 Report

Review of the manuscript entitled „Pigment production improvement using Design of Experiments 2 in Rhodotorula mucilaginosa AJB01, a red yeast isolated from Barranquilla, Colombia”

- The title of the article is very bad to read, too long and too detailed (authors should answer the question whether the country of origin of the strain is really so important that it should be included in the title ?!)

- 2.2. Molecular Identification and 3. Results: The standard procedure for genetic yeast identification involves sequencing on ITS or NL primers and this has been described. In chapter 3, the authors also provide the RPB2 gene / region. Please complete the methodology with the used primers, PCR reaction parameters and the type of polymerase used.

Why was using this region useful in identifying the tested yeast strain? The reader will not find information on this in the "Results" section. Interestingly, I did not find the chapter entitled Discussion ... in the entire manuscript. With what databases was the obtained sequence of the RPB2 region compared?

2.3. Has the method for determining carotenoids been validated? 0.8 mg of biomass was taken for determination (in my opinion it is very little ...). From the methodology, I conclude that wet biomass was collected from a petri dish? How was the dry substance content determined to give the final result?

‘AJB01 was identified 183 as Rhodotorula mucilaginosa using ITS, LSU and RPB2 regions when compared with sequences from type strains (Genbank accessions MT889967, MT889968).’ - Needs to expand the discussion, eg about what other species were similar.

Author Response

Review of the manuscript entitled „Pigment production improvement using Design of Experiments 2 in Rhodotorula mucilaginosa AJB01, a red yeast isolated from Barranquilla, Colombia”

- The title of the article is very bad to read, too long and too detailed (authors should answer the question whether the country of origin of the strain is really so important that it should be included in the title ?!)

A// We agree with reviewer. We deleted the next paragraph on title: “a red yeast isolated from Barranquilla, Colombia”

- 2.2. Molecular Identification and 3. Results: The standard procedure for genetic yeast identification involves sequencing on ITS or NL primers and this has been described. In chapter 3, the authors also provide the RPB2 gene / region. Please complete the methodology with the used primers, PCR reaction parameters and the type of polymerase used.

A// We added RPB2 comparison in our experiments to full determine the species inside complex R. mucilaginosa/alborubescens. Both regions ITS and LSU were not enough to separate the species R. mucilaginosa and R. alborubescens. In our case, we are working in a parallel experiment with the whole genome of the same strain. So, we annotated the genome using the pipeline of mycocosm (JGI genome database) and extracted the RPB2 gene from the genome. In brief, we extracted the whole genome using the Omega EZNA Fungal DNA kit, and sequenced by Novogene using Illumina platform (paired-end) with 80X of coverage. After, we trimmed the reads using FastQC and Dynamictrim, and assembled the genome using Spades. The quality of the assembled genome was assessed with Quast. The genome annotation was done using the pipeline of Mycocosm. Because the extension of this methodology, we agree in not to write all methodology to extract this gene using genome sequencing.

Why was using this region useful in identifying the tested yeast strain? The reader will not find information on this in the "Results" section. Interestingly, I did not find the chapter entitled Discussion ... in the entire manuscript. With what databases was the obtained sequence of the RPB2 region compared?

A// It´s true. We missed to add the subtitle Discussion together with results. We added the subtitle in the text.

For this clade (genus Rhodotorula), the ITS and LSU are not enough informative to separate the species R. mucilaginosa and R. alborubescens. This topology was reported by Wang et al. (2015). However, they separated these species using several genes, including RPB2. In this case, we compared our sequence with Genbank database, and the identity was 99% for R. mucilaginosa and 94% for R. alborubescens. To confirm this, we constructed a phylogenetic tree using ML with 1000 bootstrap to genus Rhodotorula. In our case, it was clear that AJB01 belong to R. mucilaginosa species. We added a supplementary Figure with the phylogenetic tree.

2.3. Has the method for determining carotenoids been validated? 0.8 mg of biomass was taken for determination (in my opinion it is very little ...). From the methodology, I conclude that wet biomass was collected from a petri dish? How was the dry substance content determined to give the final result?

A// The methodology of carotenoids extraction and quantification was proposed by Moline et al. (2012). We followed this protocol in our experiments. To determine the DWC (Dried Weight Cells), we splitted the biomass from the petri dish. A half of the biomass was used to carotenoids extraction and the other half to biomass determination. In this case, the carotenoids extraction was done using the wet biomass because the temperature or drying of biomass should modify the chemical composition of carotenoids. The tube for biomass determination was dried in a drying oven for 48h and weighted in an analytical balance. We corrected the units of standardized biomass for pigment extraction, it was 0.8 g of biomass. Also, we added a section 2.3 with methodology of biomass determination.

 ‘AJB01 was identified 183 as Rhodotorula mucilaginosa using ITS, LSU and RPB2 regions when compared with sequences from type strains (Genbank accessions MT889967, MT889968).’ - Needs to expand the discussion, eg about what other species were similar.

A// We modified the text in manuscript to explain the identification: “The ITS and LSU regions (Genbank accessions MT889967, MT889968) were compared using BLAST and matches above 99% sequence similarity with the type strain sequences. However, species level identification could not be achieved due to R. mucilaginosa and R. alborubescens have almost identical sequences on these regions. We constructed a phylogenetic tree using the RPB2 gene, classifying and confirming AJB01 as Rhodotorula mucilaginosa”

Round 2

Reviewer 2 Report

The authors made corrections, therefore I recommend the article for publication.
For the future, authors should color-mark corrections in the manuscript to make it easier for reviewers to check.